# The Role of CPNE7 (Copine-7) in Colorectal Cancer Prognosis and Metastasis

**DOI:** 10.3390/ijms242316704

**Published:** 2023-11-24

**Authors:** Hye-Jeong Kong, Dong-Hyun Kang, Tae-Sung Ahn, Kwang-Seock Kim, Tae-Wan Kim, Soo-Hyeon Lee, Dong-Woo Lee, Jae-Sung Ryu, Moo-Jun Beak

**Affiliations:** 1Department of Medical Life Science, Soonchunhyang University, Asan 31538, Republic of Korea; angelkonghj@gmail.com (H.-J.K.); kimks5005gt@gmail.com (K.-S.K.); ktwdreem@gmail.com (T.-W.K.); neekoo27@gmail.com (D.-W.L.); rjs950826@gmail.com (J.-S.R.); 2Department of Surgery, Soonchunhyang University College of Medicine, Soonchunhyang University Cheonan Hospital, Cheonan 31151, Republic of Korea; c100048@schmc.ac.kr (D.-H.K.); eyetoeye@schmc.ac.kr (T.-S.A.); marchsh93@gmail.com (S.-H.L.)

**Keywords:** colorectal cancer, *CPNE7*, siRNA, shRNA, EMT, *E-cadherin*, *N-cadherin*, *COL1A1*

## Abstract

Colorectal cancer (CRC) is one of the most common and deadly cancers in the world. However, no effective treatment for the disease has yet been found. For this reason, several studies are being carried out on the treatment of CRC. Currently, there is limited understanding of the role of *CPNE7* (copine-7) in CRC progression and metastasis. The results of this study show that *CPNE7* exerts an oncogenic effect in CRC. First, *CPNE7* was shown to be significantly up-regulated in CRC patient tissues and CRC cell lines compared to normal tissues according to IHC staining, qRT-PCR, and western blotting. Next, this study used both systems of siRNA and shRNA to suppress *CPNE*7 gene expression to check the *CPNE7* mechanism in CRC. The suppressed *CPNE7* significantly inhibited the growth of CRC cells in in vitro experiments, including migration, invasion, and semisolid agar colony-forming assay. Moreover, the modified expression of *CPNE7* led to a decrease in the levels of genes associated with epithelial–mesenchymal transition (EMT). The epithelial genes *E-cadherin* (*CDH1*) and *Collagen A1* were upregulated, and the levels of mesenchymal genes such as *N-cadherin (CDH2)*, *ZEB1*, *ZEB2*, and *SNAIL* (*SNAL1*) were downregulated after *CPNE7* inhibition. This study suggests that *CPNE7* may serve as a potential diagnostic biomarker for CRC patients.

## 1. Introduction

Worldwide, the incidence of CRC is the highest after lung cancer and prostate cancer, and the mortality rate is very high [1]. In Korea, the incidence rate is the third highest, after thyroid cancer and breast cancer [2]. Recently, the mortality rate of CRC patients has decreased due to early diagnosis including colonoscopy [3], but in about 25% of patients diagnosed with CRC, it is accompanied by metastasis [4]. Metastasis is closely related to CRC as a major cause of death in CRC patients [5]. Therefore, there is a need to identify reliable biomarkers that can be used as new targets for CRC patients. Colorectal cancer (CRC) is one of the most common malignancies worldwide and remains one of the leading causes of cancer-related death. Sustained cell proliferation and invasion, increased metastasis and drug resistance are the hallmarks of CRC. Among people diagnosed with colorectal cancer, 20% have metastatic CRC and 40% have recurrence after previously treated localized disease. The prognosis of metastatic CRC is poor, with a 5-year survival rate of less than 20%. In this context, biomarkers of metastasis are an important factor to consider, with both diagnostic and therapeutic implications. It would be essential to find non-invasive biomarkers for prognosis and early detection of CRC.

*CPNEs* (copines) are a family of membrane bound proteins that are highly conserved, soluble, ubiquitous and calcium dependent in a variety of eukaryotes [6]. The soluble membrane proteins *CPNEs* contain two tandem C2 domains at the N-terminus and an A domain at the C-terminus [7]. The diversity of *CPNEs* could have multiple roles in different signaling pathways depending on several factors, such as calcium sensitivity, lipid specificity, and target proteins [8]. *CPNE* is involved not only in the development and differentiation of the nervous system but also in the development and progression of many tumor types. *CPNE* consists of nine families—from 1 to 9—whose exact functions and biological roles are unclear, but studies have shown that the *CPNE* family may mediate signaling pathways related to tumor formation and development.

*CPNE7* has high homology with other members of the copine family, including *CPNE1*, *CPNE3*, and *CPNE6*. This copine family is a potential tumor suppressor gene [6]. *CPNE7* is a member of the *CPNE* gene family associated with cell membrane metastasis due to increased intracellular calcium [9]. It is mainly involved in calcium signaling, but it has been shown to have functions related to cell differentiation in the cytoplasm. *CPNE*7 is one of the tumor suppressor genes in breast cancer tissue. The bladder transitional cell carcinoma sequencing analysis shows the 565 candidate gene mutations that include *CPNE7* [6]. It is thought that mutation of *CPNE7* may be related to an important mechanism related to bladder cancer [10]. However, a recent study showed that high expression of *CPNE7* in mesenchymal stromal cells (MSCs) promotes oral squamous cell carcinoma (OSCC) metastasis through the *NF-κB* pathway [11]. At present, the role of *CPNE7* in cancer mechanisms is not clearly known, and therefore, further research is needed. The aim of this study is to determine the relationship between *CPNE7* expression and CRC progression and metastasis. In addition, it will prove its potential as a marker for predicting progression in CRC patients and analyze the functional effect of *CPNE7* in mediated hallmarks of cancer, including cell proliferation, migration, and invasion.

Currently, the role of *CPNE7* in cancer mechanisms is not well understood, and further research is needed. According to The Human Protein Atlas, the high expression of *CPNE7* does not show significant changes in the survival of CRC patients. However, in a group of 250 CRC patients who participated in our study, *CPNE7* confirmed its potential as a potential oncogene, and significant differences were also found in in vitro experiments. In another study based on bioinformatic analysis of transcriptome sequencing and Co-IP, *CPNE7* was shown to act as an oncogene [11]. The aim of this study is to determine the relationship between *CPNE*7 expression and CRC progression and metastasis. In addition, it will prove its potential as a prognositc marker in CRC patients and analyze the functional effect of *CPNE*7 in mediated hallmarks of cancer, including cell proliferation, migration, and invasion. It is thought that mutation of *CPNE7* may be associated with an important mechanism in bladder cancer. The role of *CPNE7* in promoting or inhibiting CRC in cancer pathogenesis is currently unclear and requires further investigation.

## 2. Results

### 2.1. Relationship of CPNE7 Expression and Clinicopathologic Factor of Patients with CRC

In previous studies, *CPNE7* was identified in the process of screening genes with significant differences in expression by NGS analysis in CRC patients [12,13,14]. The expression of *CPNE7* was confirmed via statistical analysis in approximately 250 FFPE (formalin-fixed, paraffin-embedded) tissues from CRC patients. The patient’s tissues are obtained via surgical resection of the cancer. Patients were classified according to the level of *CPNE7* expression, and age, sex, lymph node stage, meta, stage, survival rate, etc., were analyzed. The overall male/female ratio of colorectal cancer patients was 59.2% (148) male and 40.8% (102) female. Among them, 23.2% (58) of male and 27.6% (69) of female patients had high *CPNE7* expression. The percentage of patients by TNM (tumor node metastasis) stage was I stage, 14.8%; II stage, 39.6%; III stage, 59%; and 4.8% in IV stage, and the patients had 45% venous invasion and 59% lymphatic invasion. There was no statistically significant difference in the presence or absence of venous invasion and lymphatic invasion in patients with high and low expressions of *CPNE7* (Table 1). However, statistically significant results were identified in univariate (*p* = 0.032) and multivariate (*p* = 0.048) Cox regression analysis in patients with high and low expression of *CPNE7* (Table 2).

SW480, SW620, HCT116, HT29, and a colon fibroblast cell line CCD-18-co were used for qRT-PCR (Figure 1A). As a result, *CPNE7* expression was relatively higher in the colorectal cancer cell lines than in the normal cell line CCD-18-co, especially in SW480 and HCT116. This demonstrated that *CPNE7* expression was higher in the CRC cell line than in the normal cell line. Western blotting was performed using four CRC cell lines. The protein expression level of *CPNE7* was high in all CRC cell lines (Figure 1B, Appendix A). The implications of this high expression of *CPNE7* could possibly affect tumor progression and patient survival.

### 2.2. Functional Effect of CPNE7 in CRC Cell Lines

As the stage of the colorectal cancer increased, the expression of *CPNE7* was found to be statistically significantly higher. It was therefore necessary to confirm the effect of *CPNE7* expression on cell growth and proliferation. A *CPNE7*-siRNA-treated knockdown cell line generated a colorectal cancer cell line with relatively high *CPNE7* expression compared to a normal cell line. The survival rate was compared between the *CPNE7* knockdown cell line and the control cell line. This was confirmed at the mRNA and protein levels, showing that *CPNE7* expression levels were reduced by more than 80% in all four colorectal cancer cell lines. It was confirmed that SW480 (Figure 2A) at the mRNA level and SW620 (Figure 2B, Appendix A) in protein level were the most knockdown compared to the control group.

The *CPNE7* knockdown cells were used to confirm the functional changes in CRC cell lines expressing *CPNE7* (Figure 2C). As a result of comparing the difference in proliferation between the baseline CRC cell line and the *CPNE7* knockdown cell line, all four colorectal cancer cell lines showed significant differences at 24, 48, and 72 h. At 72 h, the control group grew by approximately 300%, and the *CPNE7* siRNA treated group grew by approximately 90%, which was confirmed to be a 3-fold difference. In the case of HCT116, the difference in cell growth was very large at 24 h, showing a difference of about 4 times—100% from 25%—and a difference of more than 3 times—170% from 50%—at 72 h. These data suggest that *CPNE7* expression affects the growth of colorectal cancer cells.

First, it was evaluated that the siRNA-*CPNE7*-treated cell line had decreased functionally. Next, *CPNE7* shRNA infection was performed to make a stable knockdown cell line for in vivo studies (Appendix A). mRAN and protein expression levels were decreased after *CPNE7* shRNA infection (Figure 3A,B), and the cell number in the migration and invasion assay decreased compared to the control. Proliferation was significantly lower approximately 2-fold in *CPNE7* inhibition cells, this result was similar to *CPNE7* siRNA treated CRC cells (Figure 3C). Cell migration was reduced to 65% and cell invasion to 50%. In migration, an average of 175 cells was measured in the control group, and an average of 115 was found in the *CPNE7*-shRNA-infected group. Invasion was measured in an average 146 and 76 cells (Figure 3D–G). Mean from these results, *CPNE7* is important for cell proliferation and cell migration ability in CRC.

Wound-healing experiments were performed using SW480, which had the highest expression of *CPNE7* among CRC cell lines (Figure 3H,I). It was confirmed that there was a significant difference in wound-healing ability between the control and *CPNE7* knockdown cell lines. At 12 h, approximately 34% of the wound was filled in the control group, but 25% of the wound was filled in the siRNA group, and at 24 h, it was confirmed that approximately 64% and 47% of the wound was recovered, respectively. At 36 h, the figures were 93% and 73%, respectively, confirming a 20% difference in wound healing. Finally, at 48 h, the wounds in all areas of the control group were filled first, and about 4% of the wounds in the *CPNE7*-siRNA-treated group were not filled.

From these results of functional data analysis, it was confirmed that proliferation, migration, invasion, colonization, and wound healing rate were greatly reduced after inhibition of *CPNE7* expression in all CRC cell lines, and in the same aspect as the patient data, the high expression of *CPNE7* in CRC could affect tumor progression and patient survival rate. This suggests that *CPNE7* is involved in the regulation of colon cancer cell growth in vitro. Therefore, we thought that *CPNE7* might be involved in EMT.

### 2.3. CPNE7 Inhibition Leads to the Dysregulation of EMT Signatures in CRC Cells

Given that inhibition of *CPNE7* by siRNA treatment and shRNA infection had an inhibitory effect on CRC cell motility, we sought to identify the signaling pathway of *CPNE7*. EMT enables migration and invasion of various cancers and is closely associated with poor prognosis in CRC. Among the EMT genes, we compared and confirmed the expression of eight genes: *E-cadherin*, *N-cadherin*, *MMP2*, *MMP3*, *ZEB1*, *ZEB2*, *SNAIL*, and *Collagen A1* (Figure 4). The expression of *E-cadherin* and *Collagen A1*, known as epithelial markers, increased by more than 600% and 1800%, respectively. The EMT-TFs (transcription factors) *ZEB1* and *ZEB2* increased by about 1100% and 250%, respectively, and *MMP2*, *MMP3*, and *SNAIL* were reduced by about 40%. When *CPNE7* was knocked down in CRC cell lines, the expressions of *E-cadherin* and *Collagen A1*, which are epithelial markers, increased and the expression of genes known as mesenchymal markers generally decreased, suggesting that *CPNE7* is associated with promoting EMT. This result suggests that *CPNE7* expression may be associated with EMT.

### 2.4. Patient Survival Rate According to CPNE7 Expression in FFPE Tissue in Colorectal Cancer Patients

Finally, it was necessary to verify that the function of *CPNE7* expression of in vitro displayed the same aspect in tissue samples from CRC patients (Figure 5). *CPNE7* protein expression on CRC patients was identified via IHC staining (Figure 5A). *CPNE7* was specified as a membrane protein, but IHC confirmed that it was also present in the cytoplasm and nucleus. *CPNE7* expression was higher in high-TNM-stage tissues compared to low-TNM-stage tissues. Kaplan–Meier plot analysis shows that the survival rate of patients with high *CPNE7* expression is lower than that of patients with low *CPNE7* expression (Figure 5B). At 40 months, approximately 27% of the low-*CPNE7* patients had died, whereas approximately 58% of the high-*CPNE7* patients had died. While 3 of the 123 low-expression patients died and 10 survived, at 120 months, 127 high-expression patients died and 1 survived. The *CPNE7* high/low expression patient group was classified as TNM stage 0, 1/2, 3 [15]. As a result of comparing the *CPNE7* expression level and the survival rate of patients, it was confirmed that the higher the expression of *CPNE7*, the lower the survival rate of the patients, which was statistically significant.

Schematic diagram of epithelial–mesenchymal transition pathways involving *CPNE7* in colorectal cancer (Figure 6). When CPNE7 is over-expressed in colorectal cancer, the levels of epithelial markers decrease (purple) while mesenchymal markers increase (blue).

## 3. Discussion

CRC is a disease that has been studied by many researchers but has yet to be fully conquered. Approximately 56% of patients die from this disease, and most of them are patients with metastases [6]. Therefore, early detection is very important and requires a meaningful search for a marker. The configuration, histological grade, and diameter of adenocarcinoma and lymphovascular invasion have been correlated with the degree of invasion [16]. The extension of tumor infiltration into the lymph nodes or veins has a negative impact on patient survival [17], and 50% of CRC patients still develop metastases after surgery [18]. Early diagnosis of CRC is crucial to improving the success of treatment approaches. To our knowledge, there are no reliable predictors of CRC development or progression. Therefore, early detection is very important and requires a meaningful search for markers.

In the previous study, tissue samples from 250 CRC patients were analyzed for NGS (next generation sequencing) data. There is a paper that looks at other genes selected based on the same NGS analysis [12,13,14]. Of the 250 cases examined by IHC, 127 showed some expression of *CPNE7* in CRC cells and 123 showed negative staining. In recent years, as the number of colorectal cancer patients has increased significantly and the mortality rate of colorectal cancer patients has increased, research into CRC metastasis has emerged. This research team identified a significant increase in expression in CRC patients and the role of the *CPNE7* gene, a member of the copine family associated with tumor development. *CPNE7*, a member of the copine family, consists of calcium-dependent membrane-binding proteins. There are studies showing that the increase and decrease in calcium affects human carcinogenesis [19,20] and is associated with calcium signaling [21]. In addition, there are studies suggesting that it may help prevent colorectal adenomas and reduce the risk of recurrence [22]. Deregulation of calcium sigmalling is often detrimental and has been linked to each of the ‘hallmarks of cancer’ [6]. Increased expression of inflammatory cytokines causes subsequent expression of tumor-related genes such as various target genes and *CPNE7* [23]. There is a study that Copine family affects the development and progression of the canceler. And role of *CPNE7* was tumor suppressor gene in breast cancer [6]. In another study, *CPNE7* exhibited highly contrasting expression levels between normal and tumour samples in cancer stem cells [24].

This study showed that the expression of *CPNE7* was significantly increased in the high-metastatic-stage CRC patient group. As a result of the analysis based on the patients’ TNM stage, the high expression of *CPNE7* was confirmed in the patient group with a high TNM stage, and the clinical significance of *CPNE7* expression in CRC patients was evaluated. The analysis based on the stage patients, the high expression of *CPNE7* was confirmed in the patient group with high stages (2 and 3), and the clinical significance of *CPNE7* expression in CRC patients was evaluated. This means that the presence or absence of *CPNE7* is associated with survival rate and prognosis in patients CRC malignancies. To verify this, when *CPNE7* expression was increased or decreased, changes in cell proliferation, migration, and metastasis were examined at the level of the CRC cell line to confirm whether *CPNE7* expression was actually related to the stage or metastasis of CRC. As a result of the comparative analysis of the expression of *CPNE7* in the CRC cell line and the normal colorectal epithelial cell line, the expression of *CPNE7* is higher in the CRC cell line. This confirms the same aspect as the patient data with higher expression of *CPNE7* in CRC compared to normal. To further analyze the role of *CPNE7* expression in vitro, a gene-silencing experiment was performed. Compared to the control group, the cell line proliferation, migration, and colonization abilities of *CPNE7*-suppressed cells were significantly reduced. It was confirmed that the rate of round filling was also significantly reduced. Therefore, it was confirmed that *CPNE7* expression affects tumor progression, metastasis, and survival.

There was a need to investigate which signaling pathways *CPNE7* affects in CRC survival, metastasis, and invasion, and we linked this pathway to EMT. This study has shown the change in expression of EMT-related genes from *CPNE7* expressions. Epithelial gene *E-cadherin* levels were upregulated, and mesenchymal genes, such as *N-cadherin*, were downregulated after *CPNE*7 silencing in CRC cell lines. EMT is a crucial embryonic program that is aberrantly activated in cancer and other diseases and is carried out by various EMT transcription factors (EMT-TFs). *ZEB* proteins have been shown to protect lung cancer cells from EGFR-induced senescence through their ability to down-regulate *CDKN1A* and *CDKN2B*. *CPNE7* is a transcription factor that promotes tumor invasion and metastasis by inducing EMT in CRC cell lines but also plays a dual role in suppressing it depending on the situation. Our results showed that *CPNE7* acts as a tumor oncogene in CRC cells in vitro. The expression of *CPNE7* is thought to regulate the expression of the EMT gene. A recent study showed that high expression of *CPNE7* in mesenchymal stromal cells (MSCs) promotes oral squamous cell carcinoma (OSCC) metastasis through the *NF-κB* pathway [11].

*CPNE7* mediates signaling and metastasis in many tumors. However, ‘The Human Protein Atlas’ with this classic and robust transcriptomic analysis of around 600 cases does not show any effect of *CPNE7* expression on CRC patient survival. In our study, the data from 250 colorectal cancer patients may be statistically slightly more *CPNE7* specific than the patient data from the human atlas. However, it is necessary to identify the role of *CPNE7* in other cancers as well as colorectal cancer, and the role of *CPNE7* in CRC has not been clearly identified. This study will provide important support for identifying the role of *CPNE7* in CRC. Tumor associated inflammation can induce various molecules expressed by surrounding cells or the tumor itself to create a microenvironment that is potentially pro-cancer [23]. Some studies have shown that *CPNE7* is considered a potential tumor suppressor gene in bladder cancer [6]. However, according to the results of this study, the role of *CPNE7* in CRC is different. Cell proliferation, migration, and invasion were decreased after *CPNE7* silencing in CRC cell lines (Figure 2 and Figure 3). Low expression of *E-cadherin* is associated with poor prognosis in stage III CRC [25]. Another EMT marker, *N-cadherin*, is associated with metastasis and decreased survival in CRC patients with high expression [26].

Survival of colorectal cancer patients is known to be closely related to metastasis. The aim of this study is to determine the contribution of *CPNE7* to the growth, tumorigenesis, and invasion of colorectal cancer cells, and as a result of the analysis of the patient data, it was found that the expression of *CPNE7* was associated with the metastasis of CRC. In this study, we have finally demonstrated that *CPNE7* is associated with the initiation, progression, and metastasis of colorectal cancer, and through further studies in the future, we will be able to confirm its potential as a gene therapy or biomarker for colorectal cancer patients. The ultimate goal of this research team’s ongoing research is to contribute to the treatment of CRC patients with *CPNE7* overexpression through gene silencing. As a follow-up study, we are currently synthesizing *CPNE7* antibodies and will perform in vitro and in vivo experiments using the produced antibodies. It is hoped that this will help to improve the survival rate of colorectal cancer patients.

Research on *CPNE7* in colorectal cancer with potential as a biomarker should be continuously conducted, and further verification in in vivo studies is also required. In future studies, we will generate an antibody against *CPNE7* to demonstrate its therapeutic potential targeting patients with *CPNE7* overexpression in colorectal cancer.

## 4. Materials and Methods

### 4.1. Cell Lines and Culture

The human CRC cell lines SW480, SW620, HT29, HCT116, and CCD-18-co were purchased from the Korean cell line bank (Seoul, Republic of Korea). CRC cells were grown in complete media (RPMI 1640 medium (Hyclone, Washington, DC, USA) supplemented with 10% fetal bovine serum (FBS) (Young in Frontier, Seoul, Republic of Korea) and 1% penicillin streptomycin solution (ABS) (Corning, NY, USA) at 37 °C in a humid atmosphere containing 5% CO₂. Normal cells were added to HEPES (Thermo Fisher, Waltham, MA, USA, USA15630080) 0.25% and NaHCO_3_ (Thermo Fisher, Waltham, MA, USA, 144-55-8) 0.25% in complete media.

### 4.2. Tissue Specimens from Patients

The 250 CRC specimens were obtained from patients who underwent surgery at Soonchunhyang University Cheonan hospital between 2013 and 2018. Written informed consent was obtained from all patients. They were examined for *CPNE7* expression by immunohistochemical staining. Tumor stage was evaluated according to the 7th edition of the American Joint Committee on Cancer TNM classification system.

### 4.3. Small Interfering RNA (siRNA) and Transient Transfection

The human CRC cell lines were seeded 5 × 10^5^ cells in a 10 cm cell culture dish. After 24 h, the medium was changed to a mixture of *CPNE7* siRNA, Opti MEM (Thermo fisher, Waltham, MA, USA, 31985-062), Hiperfect^®^ (QIAGEN, Hilden, Germany, 301707), which replaced the existing media. *CPNE7* siRNA was used to silence the *CPNE7* gene, and after 72 h, it was replaced with RPMI 1640 with 10% FBS and 1% ABS.

### 4.4. Short Hairpin RNA (shRNA) and Transient Transfection

Cells seeding was treated in the same way as the siRNA protocol. For transfections, cells were incubated with *CPNE7* shRNA (ORIGENE, Rockville, MD, USA, TL305242) at 50 μmol/L concentrations in Opti MEM medium. The transfection efficiencies of cell lines were tested by transfection of the shRNA plasmid negative control vector (ORIGENE, Rockville, MD, USA, TR30008). Gene knockdown was assessed by measuring mRNA by qRT-PCR 48 h after treatment.

### 4.5. Real-Time Polymerase Chain Reaction

Colon cancer cells were seeded at a density of 1 × 10^4^ cells per well in 6-well cell culture plates. After culturing for 24 h, 1 ml RiboEx (GeneAll, Seoul, Republic of Korea, 301-001) was added to each well according to manufacturer protocol. Total RNA was isolated from all cell lines using the Hybrid-R™ kit (GeneAll, Seoul, Republic of Korea, 305-101). Reverse transcription was performed using ReverTra Ace™ kit (TOYOBO, Osaka, Japan, FSQ-101). Primer of *CPNE7* (forward GTC TTC ACG GTG GAC TAC TAC T, reverse ATG CGT GTC GTA CAC CTC AAA) was used to confirm gene expression. Quantitative qRT-PCR was performed using the SYBR^®^ Green (TOYOBO, Osaka, Japan, QPK-201). The PCR cycle included one cycle at 95 °C for 1 min followed by 39 cycles at 95 °C for 15 s, 60 °C at 15 s, and 72 °C for 25 s.

### 4.6. Cell Proliferation Analysis

Cell proliferation was measured using an Ez-cytox cell viability assay kit (DOGEN, Seoul, Republic of Korea, EZ-3000). Cells were seeded in the 96-well plates. Then, at 24, 48, and 72 h after incubation, each well was treated with Ez-cytox solution. Incubation proceeded for 2 h in a humidified incubator at 37 °C containing CO_2_. The absorbance was measured at 450 nm using a Microplate reader (ThermoFisher, Waltham, MA, USA, 51119200).

### 4.7. Transwell Migration Assay

A migration assay was performed to measure the ability of cells to migrate. Using a 24-well plate containing a 6.5 mm polycarbonate filter (Corning, NY, USA, 3422), the transwell was added 250 μL of RPMI 1640 and 3 × 10⁵ cells. In the bottom chamber, 750 μL of RPMI 1640 medium containing 10% FBS and 1% ABS was added. It was incubated for 48 h in a humidified incubator at 37 °C containing CO_2_. After that, the medium in the bottom chamber and the transwell chamber was removed and the chamber was washed three times using phosphate buffered saline (PBS). To fix the cells in the membrane of the washed chamber, 4% formaldehyde was applied to the transwell and the bottom chambers and was reacted for 2 min. After washing 3 times with PBS, 100% methyl alcohol was applied and reacted for 10 min. Cells were stained by immersing the transwell chamber in 0.5% crystal violet solution for 2 min. Using an optical microscope, 5 sites of membrane were randomly photographed (900 × 1200 μm^2^).

### 4.8. Transwell Invasion Assay

Invasion was assayed using a 24-well plate containing a 6.5 mm polycarbonate filter. The inside of the transwell was covered with a non-growth factor 1:4 ratio matrigel dilution in RPMI 1640 and incubated 1 h at 37 °C. 5 × 10⁵. Cells were then plated in the insert cells, and 750 μL of RPMI 1640 medium with 10% FBS and 1% ABS was added to the bottom chamber. After 72 h of incubation, cells were fixed with 4% formaldehyde and stained with 0.005% crystal violet.

### 4.9. Semisolid Agar Colony-Forming Assay

Semisolid assays were treated with 1% agarose gel with 20% FBS in a 6 cm plate. After the bottom gel hardened, it was mixed with 0.7% agarose with 20% FBS and CRC cells. Cells were incubated for 13 days at 37 °C in a 5% CO_2_ incubator. The wells containing agarose were stained with 0.005% crystal violet, and colonies were counted.

### 4.10. Western Blot

The cells were lysed in Pro-prep™ (INtRON, Seongnam, Republic of Korea, 17081) protein extraction solution. After resting overnight, the lysates were centrifuged at 10,000× *g* for 10 min and used as the supernatant. The protein concentrations in four CRC cell lines were determined using bicinchoninic acid (BCA) (ThermoFisher, Waltham, MA, USA, 23227) assay. Equal quantities of 30 μg/μL of protein were resolved using 10% sodium dodecyl sulfate polyacrylamide gel electrophoresis. The gel was transferred onto a membrane (Millipore, Burlington, MA, USA, IPVH00010) and blocked for 1 h in 5% bovine serum albumin (BSA) (BOVOGEN, Keilor East, Australia, BSAS 0.1) in the 1× TBST (BIORAD, Hercules, CA, USA, 1706435) with 0.05% tween20 (SIGMA, St. Louis, MI, USA, 9005-64-5). The membrane incubated the *CPNE7* antibody (YNTOAB, Seongnam, Republic of Korea) diluted at 1:50 at 4 °C overnight. The anti-rabbit secondary antibody (ThermoFisher, Waltham, MA, USA, A11008) was diluted with 1:1000 after washing three times with Dulbecco’s PBS (WELGENE, Gyeongsan, Republic of Korea, LB001-02) and was reacted at room temperature for 2 h. Subsequently, the blots were detected using Pierce™ ECL Western Blotting Substrate (ThermoFisher, Waltham, MA, USA, 32209).

### 4.11. Immunohistochemistry (IHC)

Paraffin-embedded CRC tissue specimens were sectioned to a thickness of 4 μm and dried on slides for one day. The slide tissue sample was stored at 60 °C for a 1 h, and deparaffin was performed. Antigens used a 3% H_2_O_2_ at 95 °C antigen retriever buffer. For permeability, 0.2% triton solution was treated. For blocking 5% BSA in PBS was treated for 15 min, and the first antibody was treated with diluted 1:500 rabbit polyclonal antibody *CPNE7* (MYBioSourece, San Diego, CA, USA, MBS7114526) [27]. It incubated at 4 °C overnight. The second antibody was treated with diluted 1:1000 goat anti-rabbit IgG (H + L), HRP (Thermofisher, Waltham, MA, USA, 31460) for 1 h at room temperature. *CPNE7* was scored from 0 to 4 based on the IHC assay results of the FFPE block in CRC patients as read by a pathologist—0 and 1 were defined by dividing it into *CPNE7* low-expression groups and 2 and 3 were defined by division into high-expression groups [28].

### 4.12. Statistical Analysis

The Student *t*-test was performed by comparing proliferation, migration, invasion, and semisolid agar assay in a control CRC cell line and a *CPNE7* knockdown CRC cell line. ROC curve analysis was performed by comparing the survival data in high expressions of the *CPNE7* patient group and low expressions of *CPNE7* patient group. Statistical analysis was performed through the SPSS 19.0 (SPSS, Chicago, IL, USA) for Windows OS and *p* < 0.05 was considered significant. Survival data were analyzed using the Kaplan–Meier method.

## 5. Conclusions

In this study, it was discovered that *CPNE7* is a viable target gene for the treatment of CRC. The overexpression of *CPNE7* is more prevalent in patients with higher stages of CRC. Decreasing the expression of *CPNE7* in CRC significantly mitigates cancer cell functions. The upregulation of *CPNE7* in CRC might contribute to the EMT process of cancer cells. Finally, we propose that *CPNE7* serves as a predictive marker and oncogene for CRC. However, additional cytokine expression validation and in vivo experiments are necessary to confirm this hypothesis.

## Figures and Tables

**Figure 1 ijms-24-16704-f001:**
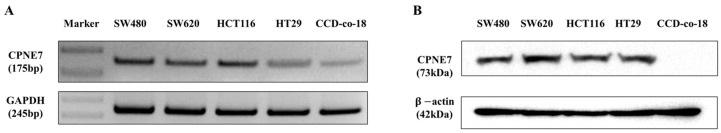
*CPNE7* expression in CRC patients’ tissue samples and CRC cell lines. (**A**) RNA-level expression of *CPNE7* (175 bp) and *GAPDH* (245 bp) in CRC cell lines. (**B**) Protein expression levels according to western blot in four CRC cell lines (SW480; colon adenocarcinoma, SW620; colorectal carcinoma, HCT116; colorectal carcinoma, HT29; colon adenocarcinoma, DLD1; colon adenocarcinoma, CCD-18-co; colon normal fibroblast).

**Figure 2 ijms-24-16704-f002:**
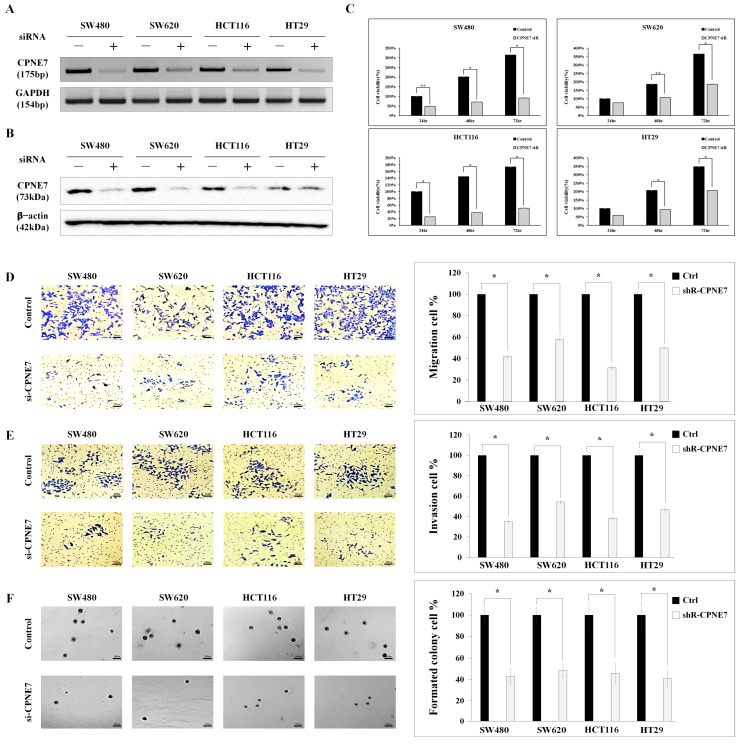
*CPNE7* inhibition by siRNA suppresses the growth of CRC cell lines. Identify the effects of *CPNE7* siRNA transfection through (**A**) qRT-PCR and (**B**) western blotting. (**C**) Cell viability test with control cells and siRNA-treated cells. Control cells and siRNA treated cells were compared through (**D**) transwell migration, (**E**) invasion, and (**F**) semisolid agar colony-forming assay (* *p* < 0.01, ** *p* < 0.05). Scale bar size 100 μm.

**Figure 3 ijms-24-16704-f003:**
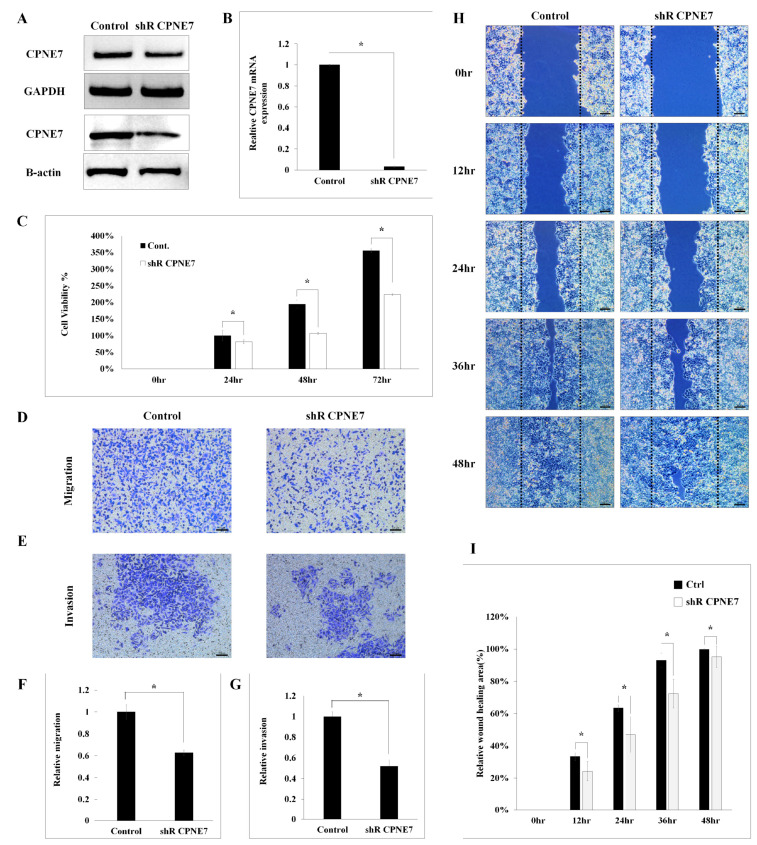
*CPNE7* inhibition by sh-*CPNE7* infection suppresses the growth of SW480 CRC cell line. The effects of *CPNE7* shRNA infection were identified through qRT-PCR (**A**,**B**) and western blotting (**A**). The (**C**) proliferation, (**D**,**F**) migration, and (**E**,**G**) invasion of *CPNE7* in the CRC cell line treated with *CPNE7* shRNA were checked. (**H**,**I**) Wound-healing assay of CRC cell lines after gene knockdown using siRNA. The black dotted line represents the first wound area of 0 hr, marked with the same position in all time zones. * *p* was considered statistically significant. (* *p* < 0.01). Scale bar size 100 μm.

**Figure 4 ijms-24-16704-f004:**
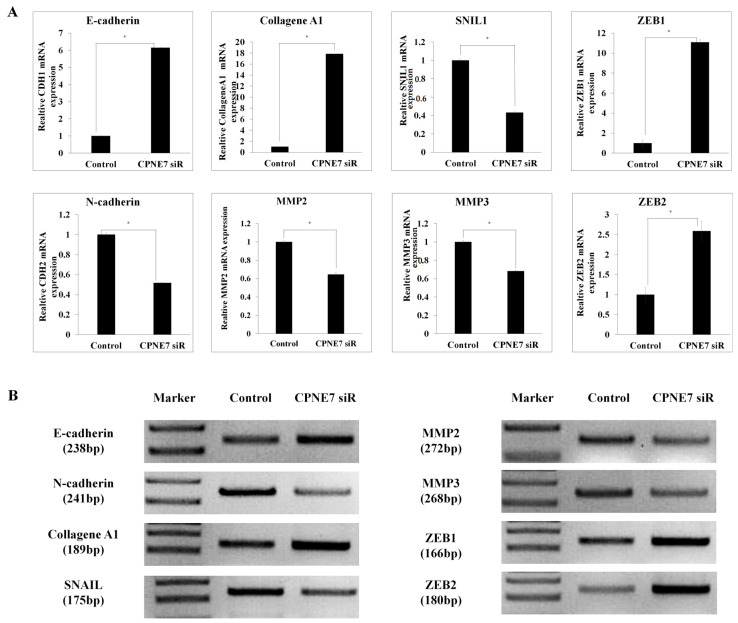
Expression of mRNA of CRC cells. (**A**,**B**) Relative mRNA expression levels of *SNAIL*, *MMP2*, *MMP3*, *Collagen A1* (*COL1A1*), *E-cadherin*, and *N-cadherin* in CRC cell line. (* *p* < 0.01).

**Figure 5 ijms-24-16704-f005:**
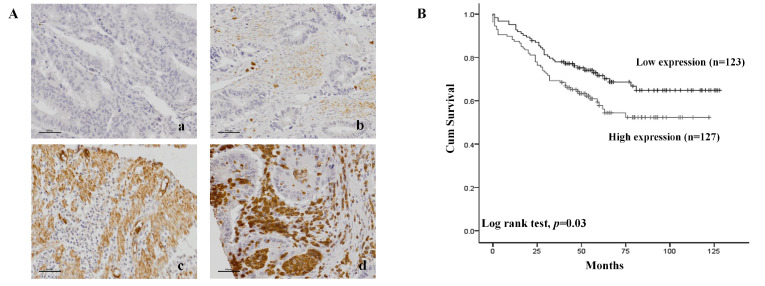
*CPNE7* protein expression level on CRC patient tissue sample according to (**A**) IHC. (**B**) The survival rate graph in high or low expression of *CPNE7* in patients. (a = score 0, b = score 1, c = score 2, d = score 3). Scale bar size 100μm.

**Figure 6 ijms-24-16704-f006:**
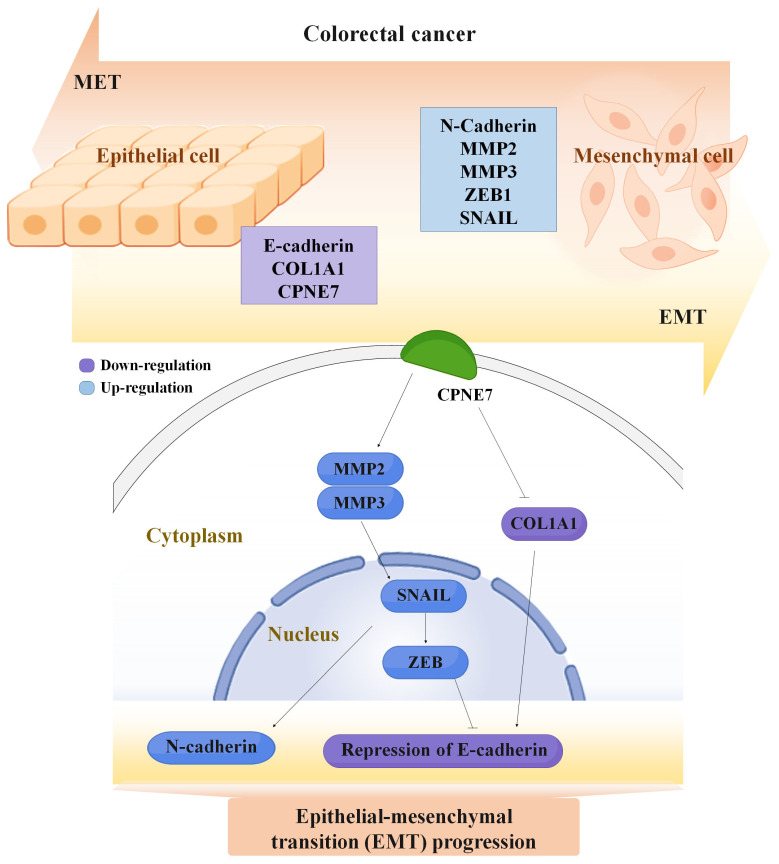
*CPNE7* study schema related to EMT. Relationship of signaling pathway at metastasis to the increase in *CPNE7*.

**Table 1 ijms-24-16704-t001:** Comparison of clinicopathologic factor and *CPNE7* expression.

Clinicopathological Factors	*CPNE7*		Total	*p* Value
	High Expression (N = 127)	Low Expression (N = 123)		
Age, years, mean (SD)	63.1 (12.6)	63.0 (12.7)	63.10 (12.6)	0.181
Sex, N (%)				0.124
Male	69 (27.6)	79 (31.61)	148 (59.2)	
Female	58 (23.2)	44 (17.6)	102 (40.8)	
TMN stage, N (%)				0.820
I	17 (6.8)	20 (8.0)	37 (14.8)	
II	52 (20.8)	47 (18.8)	99 (39.6)	
III	53 (21.2)	49 (19.6)	102 (40.8)	
IV	5 (2.0)	7 (2.8)	12 (4.8)	
Vascular invasion, N (%)				0.481
0	102 (40.8)	103 (41.2)	205 (82.0)	
1	25 (10.0)	20 (8.0)	45 (18.0)	
Lymphatic invasion, N (%)				0.230
0	93 (37.2)	98 (39.2)	191 (76.4)	
1	34 (13.6)	25 (10.0)	59 (23.6)	

**Table 2 ijms-24-16704-t002:** Cox regression analysis of the clinicopathological factor (* *p* < 0.05).

Clinicopathological Factors	Multivariate Analysis		Univariate Analysis	
	Hazard Ratio (95% CI)	*p*	Hazard Ratio (95% CI)	*p*
Age, years, mean (SD)	1.63(1.00–2.68)	0.050	1.51 (0.94–2.42)	0.087
Sex, N (%)				
Female vs. Male	0.98 (0.64–1.51)	0.952	1.13 (0.74–1.72)	0.548
TNM stage, N (%)		0.005 *		0.001 *
I	1.00		1.00	
II	1.08 (0.50–2.30)	0.838	1.16 (0.54–2.26)	0.692
III	1.90 (0.91–3.95)	0.085	2.01 (0.98–4.13)	0.055
IV	4.05 (1.48–11.09)	0.006 *	4.60 (1.77–11.96)	0.002 *
Venous Invasion				
High vs. Low	0.87 (0.41–1.85)	0.735	1.59 (0.97–2.59)	0.062
Lymphatic Invasion				
High vs. Low	1.74 (0.89–3.37)	0.102	1.94 (1.25–3.03)	0.003 *
*CPNE7* expression				
High vs. Low	1.54 (1.00–2.38)	0.048 *	1.58 (1.03–2.41)	0.032 *

## Data Availability

Data will be made available on request.

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
