# Peer review of "The Role of CPNE7 (Copine-7) in Colorectal Cancer Prognosis and Metastasis"

_ijms, 2023, doi:10.3390/ijms242316704_

Round 1

Reviewer 1 Report (Previous Reviewer 2)

Comments and Suggestions for Authors

The manuscript can be accepted in its present form.

Author Response

Thank you for revised.

Reviewer 2 Report (Previous Reviewer 3)

Comments and Suggestions for Authors

I have no objection.

Author Response

Thank you for revised.

Reviewer 3 Report (New Reviewer)

Comments and Suggestions for Authors

The authors were targeting a promising new aspect for treatments in carcinomas, however, there are some parts which need to be improved in order to make their message clearer. 

The MS's weakest part is the results section, mostly because of the presentation of the data. The quality overall in all the figures is way too weak. Thus I was unable to read those graphs, even with a greater magnification the quality just got worse. In order to make a proper review we do need a better quality of the figures.

Please check out the word template settings for the word splits. Some parts of the MS contain mistakes in the setup.

Reading through the manuscript the following typos, comments I do have:

Introduction:
However, no effective treatment has yet been found. grammar wise unpleasant

Line 18 in vitro should be in italics

Line 19 EMT does not have a gene expression, it is a process, which means several genes are involved in that process

Line 24 Keywords: it would be beneficial to involve the important genes which are the target of the MS

Line 33 : bi-omarkers strange split

Line 57: the statement is strong, there must be a reference for that

Line 65 the sentence is strange here as well, patients. And ? 

Line 68, this sentence was repeated third times, in a different phrase, i do think it is not needed here again

Line 72 in vitro should be in italic

Line 74 there is missing space between the text and the reference [12]

Again the aim of the study is repeating itself

Line 74-81 : these sentences do not make any sense, the way they are repeating the same "not been adequately investigated " = "further research is needed" The authors should revised this part carefully, to avoid the repetation

Line 89 there is missing space between the text and the reference []

Line 90 what is an FFPE tissue, abbreviation meaning is missing

Line 104 space is missing before the ref

Line 108 space is missing before the ref

Line 107 western blot W should be uppercase

Line 116 Space is missing before the ()

Figure 2. panel A western blots , the blot is really weird, HT29 cells pair before that it look like the blot was merged together, can the authors elaborate in this ? 

I'm not sure about that panel A is qPCR result!?

Panel C axis labelling and quality of the figures are weak

Figure 3. panel C where are the SD or SEM values from the graph? I see that it must be a relative figure, but still, I guess the authors did not make the experiment only ones.

Figure 5 is not easy to understand, still don't know where the a,b,c,d parts are related , I assume that is the stages?

Line 414 typo WESTERN BLOT

Comments on the Quality of English Language

Quality of the English is okay, minor changes needs to be done. 

Author Response

Thank you for revised.

1. I modified the grammar of the sentence.
[However, no effective treatment for the disease has yet been found.]

2. All of the plaintiff's 'in-vivo, in-vitro' are in italics.

3. The sentence has been modified to mean 'multiple genes involved in EMT’.
[Moreover, the modified expression of CPNE7 led to a decrease in the levels of genes associated with epithelial-mesenchymal transition (EMT).]

4. Keywords included important genes that are the main targets in the manuscript.
[E-cadherin, N-cadherin, COL1A1 ]

5. Reference has been added.

6. The sentence of the manuscript has been revised.
[The role of CPNE7 in promoting or inhibiting CRC in cancer pathogenesis is currently un-clear and requires further investigation.]

7. We added abbreviations for FFPE and reviewed the abbreviations that were omitted from the manuscript.

8. We improved the image quality of Figure 2 and replaced the qPCR picture of 2-A with a better one.

And in Figure 2-B, the band of HT29 in the Westren blot picture showed significant statistics in gene expression as a result of measuring intensity.

9. a,b,c,d in Figure 5-A are representative slides of scores 1,2,3,4 read by professional pathologists. The description has been added to the manuscript.

Reviewer 4 Report (New Reviewer)

Comments and Suggestions for Authors

The authors give an new method baced on the role of CPNE7 (copine-7) in CRC progression and metastasis,  for Colorectal cancer (CRC). The results of this study show that CPNE7 exerts an oncogenic effect in CRC, as the epithelial genes E-cadherin (CDH1) and Col-lagene A1 were upregulated and the levels of mesenchymal genes such as N-cadherin, ZEB1, ZEB2 and SNAIL (SNAL1) were downregulated after CPNE7 inhibition.

The study is written in standart English, structurated well and inovative - suggests that CPNE7 may serve as a potential diagnostic biomarker for CRC patients.

Minor recomendations :

1.Table 1- too small and blurry; need to be done in the template;

2. row 96........stage was â…  stage 14.8%, â…¡  stage 39.6%, â…¢ stage 59% and 4.8% in â…£ stage. (Roman numerals again)

3.Table 2- too small and blurry; need to be done in the template; p<0.05 vs......

4. it might be better to separate table 2 and figure 1 with text;

5. figure 2 C .... the figures are cut out and it shows; to separate

6. Figure 4 c part - recoended to be as separate, as sheme 1 

7. conclusions is too small; it is necessary the authors to add a limitation of the study and future experiments;

8. are these the possible articles to support the research; to add articles from 22-23 years

Comments on the Quality of English Language

Minor editing of English language required

Author Response

Thank you for revised.

  1. I have improved the image quality of the plaintiff's Table.
  2. I reviewed the Roman numerals of 96line, but we didn't find an error. Thank you.
  3. We improved the quality of the manuscript's table and added text to make it easier to distinguish between table 2 and Figure 1.
  4. The figures in Figure 2-C have been added to the contents of the manuscript.
  5. We improved the image quality of Figure 2.
  6. According to the recommendations, Figure 4-C is classified as a sheet and as Figure 5.
  7. Conclusion was added for research and future experiments. Conclusion was added for research and future experiments.
  8. Reference has been added to the manuscript.

Reviewer 5 Report (New Reviewer)

Comments and Suggestions for Authors

Thank you for the interesting research. The topic is fascinating and the research well conducted.  I think authors should put the Material and Methods section after the introduction and not before the conclusion

Author Response

Thank you for revised.

On the template of the manuscript provided by MDPIijms, it was confirmed that the 'material and methods' part is located after the discussion.

If we are mistaken, we will correct it.

Reviewer 6 Report (New Reviewer)

Comments and Suggestions for Authors

This manuscript aims to determine the relationship between CPNE7 expression and CRC progression and metastasis. 

Please consider the following suggestions:

Please insert the definition of each abbreviation used in the manuscript.

Please expand the Conclusions section and refer to the results presented in this manuscript.

Please insert more references published more recently.

Author Response

Thank you for revised.

We reviewed the abbreviations that were omitted from the manuscript.

Conclusion was added for research and future experiments. Conclusion was added for research and future experiments.

Reference has been added to the manuscript.

Round 2

Reviewer 3 Report (New Reviewer)

Comments and Suggestions for Authors

The authors have been changing some parts of the manuscripts, my previously mentioned problem was the quality of the figures, however the problem still exists. If somebody wants to print the article, or even wants to take a closer look at the figures, it has a really weak quality. Please do zoom in the figures and you will see that the DPI values are not even reaching the 300 dpi, which should be the minimum resolution in any kind of figures. Until that, the figures are just blurred black and white columns. 

The material methods part Western blot part still has the typo as "Wesern blot", please do take care of the spelling as we are talking about a scientific paper. 

Author Response

Thank you for revised.

We found that 'picture image compression' was applied to the option in the process of storing words.The quality of the figures has been improved by improving picture quality by checking them.

Additional typos have been corrected. 'western blot'

Thank you.

Round 3

Reviewer 3 Report (New Reviewer)

Comments and Suggestions for Authors

Hereby I do accept the changes and the current form of the MS. 

This manuscript is a resubmission of an earlier submission. The following is a list of the peer review reports and author responses from that submission.

Round 1

Reviewer 1 Report

Comments and Suggestions for Authors

In this manuscript authors present data suggesting an over-expression of CPNE7 in colorectal cancer in relation with EMT and metastasis. Experiments are well performed and clearly explained. However, too many information lack and some complementary experiments are need to be convincing (see below). English should be extensively reviewed as well.

Concerning CPNE7 expression, the authors mention a previous NGS study (page 7 line 155) but without any reference of publication. They did not detail results of this first investigation. If we refer to Human Protein Atlas, low and high transcription of CPNE7 have no incidence on survival. How authors interpret that ? They should at least comment their data in comparison to those from open bases.

In terms of protein expression in CRC, all data result from an IHC study. However, the methods for quantifying and defining high and low expression was not explained. Moreover, a polyclonal antibody was used, with all the risks of unspecific binding of a polyclonal batch. Therefore staining at list on few samples should be confirmed by another primary antibody, preferentially a monoclonal one. Regarding the staining, image in Figure 1A should be enlarged to be really visible. Their staining should also be described in Results and not in Discussion. Surprisingly, in this study expression was observed mainly in nucleus and cytoplasm (page 7 line 169). This is not discussed by authors while CPNE7 is a membranous protein. In was not comment as well whether expression was detected in cancer cells or in TME. This is important because authors consider their results similar those from OSCC (page 7 line 193) where CPNE7 was detected in MSCs while they performed all their experiments with siRNA and shRNA on cancer cells.

The high expression in CRC cell lines should be also more defined. High expression in comparison to what ? Non-CRC Cell lines, ideally cells from normal colon should be tested by Western-blot to establish a comparison.

Concerning experiments with siRNA and ShRNA, their sequences or at least their origins should be detailed. As with the antibody for IHC study, a second siRNA should be initially tested for confirming data with the previous one. Figures showing effects on proliferation and consequently on colony formation are clear but data should be expressed differentially for migration and invasion, i.e. in % of the total cells.

Finally, authors suggest in Abstract that CPNE7 could be a target for CRC treatment but they did not discussion any option of treatment targeting CPNE7. At this step, it would be more reasonable to propose CPNE7 as biomarker only.

Comments on the Quality of English Language

As mentioned previously, some sentences are really difficult to read ... 

Author Response

  1. Concerning CPNE7 expression, the authors mention a previous NGS study (page 7 line 155) but without any reference of publication.
    : For prior study to explore genes related to the progression of CRC, an NGS analysis was conducted on 250 colorectal cancer patients. No paper has been published based on prior research in that paper. Instead, there is a paper that studies other genes selected based on the above NGS analysis[17,18,19,20,21].
  2. They did not detail results of this first investigation. If we refer to Human Protein Atlas, low and high transcription of CPNE7 have no incidence on survival. How authors interpret that ? They should at least comment their data in comparison to those from open bases.
    : Among the reference papers, there are results of migration and invasion reduction when CPNE7 is suppressed and migration and invasion promotion experiments when CPNE7 is overexpressed in oral squamous cell carcinoma. The paper was suggested that CPNE7, which was also expressed promotes the progression of CRC by activating the MAPK signaling pathway[12].
  3. In terms of protein expression in CRC, all data result from an IHC study. However, the methods for quantifying and defining high and low expression was not explained.
    : We scored it as score 0,1,2,3 through IHC assay results of FFPE block in CRC patients, 0 and 1 were defined by dividing it into CPNE7 low-expression groups, and 2 and 3 into high-expression groups. Among the references, there are cases in which 0 and 1 are divided into low groups and 2 and 3 are divided into high groups according to the degree of IHC dyeing in CRC patients[13].

  4. Moreover, a polyclonal antibody was used, with all the risks of unspecific binding of a polyclonal batch. Therefore staining at list on few samples should be confirmed by another primary antibody, preferentially a monoclonal one. Regarding the staining, image in Figure 1A should be enlarged to be really visible. Their staining should also be described in Results and not in Discussion.
    : Since there is no CPNE7 antibody currently available, I attached the experimental results demonstrated by RNA level through qRT-PCR using CPNE7 primer. Also, if we can extend the revision period further, we will submit Western results sincerely. And we revised contents of IHC figure size and contents.

  5. Surprisingly, in this study expression was observed mainly in nucleus and cytoplasm (page 7 line 169). This is not discussed by authors while CPNE7 is a membranous protein. In was not comment as well whether expression was detected in cancer cells or in TME. This is important because authors consider their results similar those from OSCC (page 7 line 193) where CPNE7 was detected in MSCs while they performed all their experiments with siRNA and shRNA on cancer cells.
    : The CPNE7's IHC is also dyed in the nucleus and cytoplasm, and there are papers and human protein atlas that support it[13]. Currently, there are no CPNE7 antibodies available in lab, and it takes more than two months to order and receive them. Therefore, it is difficult situation to conduct IHC re-experiments. If you extend the revision period, I will submit the results of the experiment you want.

  6. The high expression in CRC cell lines should be also more defined. High expression in comparison to what ? Non-CRC Cell lines, ideally cells from normal colon should be tested by Western-blot to establish a comparison.
    : For comparative analysis of CPNE7 gene expression in CRC cell line was analyzed based on the normal cell line CCD-18-co. We added PCR data with CCD-18-co cell line.

  7. Concerning experiments with siRNA and ShRNA, their sequences or at least their origins should be detailed.
    : We added the corresponding content to this paper.

  8. As with the antibody for IHC study, a second siRNA should be initially tested for confirming data with the previous one.
    : And we revised deployment order of IHC contents.

  9. Figures showing effects on proliferation and consequently on colony formation are clear but data should be expressed differentially for migration and invasion, i.e. in % of the total cells.
    : We revised the figures of (fig2C,D,E) figures and contents of the paper to %.

  10. Finally, authors suggest in Abstract that CPNE7 could be a target for CRC treatment but they did not discussion any option of treatment targeting CPNE7. At this step, it would be more reasonable to propose CPNE7 as biomarker only.
    :We revised and supplemented abstract of the paper.

Reviewer 2 Report

Comments and Suggestions for Authors

In this paper, the authors emphasize the significance of early detection and meaningful markers for colorectal cancer (CRC), given its high mortality rate and frequent occurrence of metastasis. They present data and correlations between the configuration, histologic grade, and diameter of adenocarcinoma, as well as lymphovascular invasion, with the level of invasion, underscoring the adverse impact of tumor infiltration on patient survival. CPNE7 expression is observed in CRC cells and is linked to calcium signaling and human carcinogenesis. The study suggests that CPNE7 expression significantly increases in CRC patients with advanced metastasis, potentially serving as a marker for survival and metastasis. Suppression of CPNE7 leads to reduced cell proliferation, migration, and invasion, while influencing the expression of EMT-related genes such as E-cadherin, N-cadherin, and SNAIL. The study also indicates a potential role for CPNE7 in cancer metastasis in CRC and oral squamous cell carcinoma (OSCC).

Please address these points:

How does CPNE7's role in CRC differ from its role in bladder cancer, where it is considered a potential tumor suppressor gene?

How does the alteration of EMT-related gene expression, such as upregulation of E-cadherin and downregulation of N-cadherin and SNAIL, contribute to the effects observed after CPNE7 suppression in CRC cell lines?

Author Response

  1. How does CPNE7's role in CRC differ from its role in bladder cancer, where it is considered a potential tumor suppressor gene?
    : Among the reference papers, CPNE7 was found to have non-synonymous mutation in the bladder canceler and to have no other specific genetic mutation. [10]. And other paper was suggested that CPNE7 promotes  proliferation through and MAPK signaling pathway in CRC[12].

  2. How does the alteration of EMT-related gene expression, such as upregulation of E-cadherin and downregulation of N-cadherin and SNAIL, contribute to the effects observed after CPNE7 suppression in CRC cell lines?
    : Among the reference papers, there are results of migration and invasion reduction when CPNE7 is suppressed and migration and invasion promotion experiments when CPNE7 is overexpressed. The paper were showed E-cadherin, N-cadherin western blot results to verify EMT promotion/inhibition[12].

Reviewer 3 Report

Comments and Suggestions for Authors

Topic of manuscript is interesting, actual, important and suitable for IJMS. Nevertheless some point can be taken for the improvement of manuscript.

Some abbreviation such as TNM are not defined.

Results of in vitro studies should summarized be in table.

Legend of Figure 3. Term of motility is probably typo.

Difference between CPNE7 effect and migration and invasion could suggest, that CPNE7 could expression of matrix metalloproteases, most probably via NF-kB signalling. Is anything known about this topic?

Selection of patients should be described more detailed.

In this case of in vitro test should showed more experimental details such as, citation of protocol, used amount of shRNA, manufacturer/shRNA preparation protocol, number of replicants, in this case of western blot, every measured blot should be presented in the SI.

Subsection 4.10 title did not match the title does not correspond to the content.

Author Response

  1. Some abbreviation such as TNM are not defined.
    : We revised abbreviation in paper.
  2. Results of in vitro studies should summarized be in table. 
    : Which in-vitro results do you mean exactly among the various figures written in the paper. If you tell me what form I should summarize, I will fill it out sincerely and send it to you.
  3. Legend of Figure 3. Term of motility is probably typo. 
    : We modified the words in the paper.
  4. Difference between CPNE7 effect and migration and invasion could suggest, that CPNE7 could expression of matrix metalloproteases, most probably via NF-kB signalling. Is anything known about this topic?
    : Among the reference paper, the CPNE7 was facilitate the progression of CRC by interacting with PKM2 and initiating the MAPK signal pathway[12].

  5. Selection of patients should be described more detailed.
    : We scored it as score 0,1,2,3 through IHC assay results of FFPE block in CRC patients, 0 and 1 were defined by dividing it into CPNE7 low-expression groups, and 2 and 3 into high-expression groups.

  6. In this case of in vitro test should showed more experimental details such as, citation of protocol, used amount of shRNA, manufacturer/shRNA preparation protocol, number of replicants, in this case of western blot, every measured blot should be presented in the SI.
    : We modified the contents of Materials and Methods in the paper. Currently, there are no CPNE7 antibodies available in lab, and it takes more than two months to order and receive them. Therefore, it is difficult situation to conduct Western re-experiments. If you extend the revision period, I will submit the results of the experiment you want.

  7.  Subsection 4.10 title did not match the title does not correspond to the content.
    : We modified the title of Materials and Methods in the paper.

Reviewer 4 Report

Comments and Suggestions for Authors

The authors did excellent work RNAi mediated work. Please change good figures 2 D, E, F. It's unclear. Then rewrite the discussion part and conclusion. It is lacking some results. 

Author Response

  1. The authors did excellent work RNAi mediated work. Please change good figures 2 D, E, F.
    : We modified the figure2 D,E,F in the paper.

  2. It's unclear. Then rewrite the discussion part and conclusion. It is lacking some results.
    : We modified the discussion, conclusion and result in the paper.

Round 2

Reviewer 1 Report

Comments and Suggestions for Authors

The authors replied to some questions in this version, however, the main questions remain on table:

They are not so many studies referring to CPNE7 expression in CRC. It is therefore very important to send any information that could reinforce the message of this manuscript. Indeed, it is difficult to mention a NGS analysis on 250 CRC patients without any reference. If this work was not published yet, data should be summarized in this paper, at least as Sup. Data.

The fact that Human Protein Atlas with this classical and robust transcriptomic analysis of around 600 cases does not show any effect of CPNE7 expression on survival of CRC patient remains questionable. The authors have to mention this data, discuss it and propose some explanation for that one can consider as a discrepancy. The fact that some papers show a decrease of migration and invasion when CPNE7 is suppressed is different that such a large prospective study and lead to conclusion that are not totally comparable.

Verification of the protein expression by another anti-CPNE7 antibody is crucial here and I support an extension of the revised period for confirming this major point. Cytoplasmic and nucleus expression have to be explained with at least some hypothesis. The more obvious conclusion at this step is that anti-CPNE7 antibodies used are not specific enough. If tools used by HPA for measuring the transcription are robust, it is not the case concerning protein expression: this data base refers to published data that result from experiments with sometimes doubtable quality antibodies on fixed tissues.

Low and high expression that support this work remains to be defined and write in materials and methods. Even if a H score is not calculated, scoring by IHC is not an approximate estimation but should be supported by grading staining intensity, % of positive cells and so on.

Remark: For a faster reviewing, it is more convenient to review a second version with revision marks…

Several typos to be checked (ex: 4.10 Western-blot)

Comments on the Quality of English Language

No major comment

Author Response

Thank you for revision.

In this paper, we mentioned the data that human protein atlas with approximately 600 classical and robust transcriptional analyses do not show the effect of CPNE7 expression on the survival of CRC patients and add an explanation to this.

NGS analysis data from 250 colorectal cancer patients have been used in several previous studies (ref17,18,19,20,21).

We added to the materials and methods the definitions of low and high expressions.
For score of IHC was read by a pathologist according to the staining intensity.
And in the case of cytoplasm and nuclear expression of CPNE7, it was confirmed that it was strongly expressed in the cell membrane by referring to Protein atlas and papers, but nuclear and cytoplasm were also expressed.
Please refer to the link below.
https://www.proteinatlas.org/ENSG00000178773-CPNE7/tissue/colon#imid_11539049

For Western blot,
The final purpose of this research team's ongoing research is to contribute to the treatment of CRC patients with CPNE7 overexpression through gene suppression.
Currently, we are synthesizing CPNE7 antibodies as a follow-up study of the paper submitted to MDPI iJMS, and we will conduct in vitro and in vivo experiments through the manufactured antibodies. 
Antibodies are expected to be produced around October, and we have identified several types of clones that work through clone screening.
We contacted the antibody manufacturer and confirmed that we could receive the test antibody pre-made for testing.
The attached PDF is the result of Western blot with several types of clones that work with the clone screening and production process that we did to produce antibodies.
Through the above description, thankfully, the experiment with antibodies has been approved.
And we successfully added the new Western blot result. I will attach the PDF of the figure.

Since I can only do one attached file, I am sending you a combination of Western blot and CPNE7 antibody.

Reviewer 3 Report

Comments and Suggestions for Authors

I have few only objection 

point 2. I would like recomend presented value represent CPNE7 effect on the proliferation,  migration and  invasion.

In this RNA

please add reference for protocol

add information about used amount and sequences

Comments on the Quality of English Language

The quality of the English language and clarity of the manuscript is acceptable for publication.

Author Response

We added the protocol information about used amount and sequences this paper.

Round 3

Reviewer 1 Report

Comments and Suggestions for Authors

Some comments:

1/ If the NGS was already published in ref 17-21, why where is still no reference in sentence line 82. (by the way,English of this sentence should be revised)

2/ Definition of staining should be defined with rigor:  low, high, medium and so on is subjective and could differ from a reader to another one. Is it based on staining intensity ? In this case, is the  percentage of positive cells considered as well ? None of biomarkers is evaluated with a so vague definition. (by the way, a typo was previously mentioned in 4.10 "Western-blot" and not "Wesern bolt", I supposed that the suggestion to check typos was not follow ... )

3/ As previously mentioned, Human Protein Atlas is a good reference for RNA transcription but not for protein expression. The reason is that data stocked in HPA regarding proteins come from publications without any additive control. If  antibodies used are not really specific some nuclear or cytoplasmic expression could be mentioned by error. If you use the same antibody, you will just reproduce this error. For this reason, it is important to refer not only to HPA. but to publications by specifying tested antibodies. Obviously, the same observation with several different antibodies could reinforce this information. And, finally, concerning this question, if nuclear and cytoplasmic expression could be considered as confirmed, the authors could propose reason or role of CPNE7 expression in these loci. 

To go back to Human Protein Atlas, as previously mentioned, based on data presented on this site, high expression of CPNE7 does not show significant variation on survival of patient with CRC. I did not see any comment or discussion of this point in the manuscript, while requested. A version with marked revision remains extremely useful for a rapid revision ... 

4/ Good to know that monoclonal antibodies will be available soon to confirm experiments. Through the response of authors, I understand that they plane to rather use this/ese new antibody/ies for a follow-up study. If it is the case and if only IHC data with polyclonal antibody are presented, it is extremely important to add many reservations concerning these results and to clearly specify in perspective that protein expression in CRC remains to be confirmed by complementary experiments. However, the better option is to include some controls with one of this new antibody, and to validate some staining.

Comments on the Quality of English Language

English should be rechecked.

Author Response

Thank you for revision.

We marked the revised manuscript in red for rapid revision.

  1. We modified the sentence in line 82 and added a reference.
  2. Added information about IHC dyeing definition to 4.11.

   The high/low expression patient group of CPNE7 was classified as TNM stage 0,1/2,3[13]

   The patient's Tissue written on the IHC is obtained only cancer by surgical operation.

   So we didn't consider the percentage of positive cells.

   I'm sorry that I missed something about the typo correction.

   We revised the title of 4.10

  1. Information on CPNE7 antibodies used to confirm protein expression was added and specified in 4.10.

   Also, I added several papers with CPNE7 protein dyed as a reference, so please check it.

   As you suggested, We added a sentence related to the expression of CPNE7 to the manuscript.

  1. Added information on CPNE7 antibodies used to confirm protein expression to 4.10.

   And after completing the antibody, the contents of the experiment to be conducted as a follow-up study were briefly added to the discussion.

   We considered the better options you provided, but after checking the availability of the patient's tissue, IHC re-experimenting is unlikely.

Thank you.
